# The Role of Trace Elements in COPD: Pathogenetic Mechanisms and Therapeutic Potential of Zinc, Iron, Magnesium, Selenium, Manganese, Copper, and Calcium

**DOI:** 10.3390/nu16234118

**Published:** 2024-11-28

**Authors:** Mónika Fekete, Andrea Lehoczki, Tamás Csípő, Vince Fazekas-Pongor, Ágnes Szappanos, Dávid Major, Noémi Mózes, Norbert Dósa, János Tamás Varga

**Affiliations:** 1Institute of Preventive Medicine and Public Health, Faculty of Medicine, Semmelweis University, 1089 Budapest, Hungary; fekete.monika@semmelweis.hu (M.F.); ceglediandi@freemail.hu (A.L.); csipo.tamas@semmelweis.hu (T.C.); pongor.vince@semmelweis.hu (V.F.-P.); major.david@semmelweis.hu (D.M.); mozes.noemi@semmelweis.hu (N.M.); dosa.norbert@semmelweis.hu (N.D.); 2Health Sciences Program, Doctoral College, Semmelweis University, 1085 Budapest, Hungary; 3Heart and Vascular Center, Semmelweis University, 1122 Budapest, Hungary; drszappanos@gmail.com; 4Department of Rheumatology and Clinical Immunology, Semmelweis University, 1088 Budapest, Hungary; 5Department of Pulmonology, Semmelweis University, 1083 Budapest, Hungary

**Keywords:** COPD, oxidative stress, zinc, copper, iron, magnesium, manganese, selenium, calcium, trace elements, inflammation

## Abstract

Background: Chronic obstructive pulmonary disease (COPD) is a progressive, inflammatory airway disorder characterized by a gradual decline in lung function and increased oxidative stress. Both oxidative stress and inflammation are central to its pathophysiology, with trace elements such as zinc, copper, iron, manganese, magnesium, selenium, and calcium playing key roles in various cellular processes. Objective: This article reviews the role of trace elements in COPD, focusing on their involvement in disease pathogenesis and their therapeutic potential. Specifically, we examine the effects of zinc, copper, iron, magnesium, manganese, selenium, and calcium in COPD. Methods: We performed a comprehensive narrative review of the literature across databases including PubMed, Web of Science, Cochrane Library, and Google Scholar, identifying studies that explore the therapeutic effects of trace elements in COPD. The studies included in the review consisted of cohort analyses, randomized controlled trials, and clinical investigations. Results: Zinc, copper, iron, magnesium, manganese, selenium, and calcium are critical to both the pathophysiology and management of COPD. These trace elements contribute to the regulation of inflammation, the modulation of oxidative stress, and the maintenance of lung function. Zinc and copper, for instance, reduce oxidative stress and modulate immune responses, while iron is essential for oxygen transport. Magnesium, manganese, selenium, and calcium are vital for muscle function, respiratory performance, reducing inflammation, and improving pulmonary function. Conclusions: The minerals zinc, copper, iron, magnesium, manganese, selenium, and calcium may contribute to beneficial effects as part of the standard therapeutic management of COPD. Maintaining optimal levels of these trace elements may support the regulation of inflammatory processes, a reduction in oxidative stress, and an improvement in the pulmonary function. However, further clinical research is necessary to confirm their efficacy and establish safe dosage recommendations in COPD treatment.

## 1. Introduction

Chronic obstructive pulmonary disease (COPD) is one of the leading causes of morbidity and mortality worldwide [1,2]. According to the Global Initiative for Chronic Obstructive Lung Disease (GOLD), COPD is a progressive disease characterized by persistent airflow limitation and a heightened chronic inflammatory response in the airways and lungs to harmful particles or gases [3,4]. By 2030, COPD is projected to become the third leading cause of death globally [5].

In COPD, the consequences of the intense inflammatory response include increased mucus production (chronic bronchitis), tissue destruction (emphysema), and the disruption of normal repair and defense mechanisms, which causes inflammation and scarring of the small airways [6,7,8,9,10]. As highlighted by Barnes, in addition to inflammation, two other critical processes contribute to the pathogenesis of COPD: an imbalance between proteases and antiproteases and a disrupted oxidant–antioxidant balance, leading to oxidative stress in the lungs [11]. Oxidative stress not only affects the severity of the disease, but also its management. As a result, antiproteinases become inactivated, airway epithelial cells are damaged, mucus production is enhanced, neutrophil accumulation increases in the pulmonary microvasculature, and the gene expression of inflammatory mediators is upregulated [12]. Antioxidant defense systems play a key role in counteracting the effects of oxidative stress [13].

Elevated levels of reactive oxygen species (ROS), such as superoxide, hydroxyl radicals, and hydrogen peroxide, can inflict significant cellular damage by oxidizing key biomolecules, including deoxyribonucleic acid (DNA), proteins, and lipids. Beyond immediate damage, excessive ROS levels and oxidative-stress-mediated macromolecular damage contribute to the accelerated onset of cellular and tissue aging phenotypes [14,15], such as telomere shortening [16,17], mitochondrial dysfunction [18,19], necroptosis [20], and cellular senescence [21,22,23,24,25,26]. Oxidative stress resilience pathways are particularly critical in the hyperoxic environment of the lungs, where cells are exposed to higher oxygen levels than in most other tissues [27,28,29]. This elevated oxygen availability predisposes pulmonary tissues to oxidative damage due to an increased likelihood of ROS generation. Antioxidant defenses, including enzymatic systems such as superoxide dismutase, catalase, and glutathione peroxidase, work to neutralize these reactive molecules and maintain cellular homeostasis [27,28]. When these systems are overwhelmed or impaired, the lungs become vulnerable to oxidative injury, inflammation, and structural damage, all of which are hallmarks of COPD pathology. Additionally, impaired cellular stress resilience, including diminished antioxidant capabilities, amplifies the deleterious impact of ROS, perpetuating a cycle of oxidative damage and deteriorating tissue function [27,28,30]. Aging itself is associated with impaired stress resilience [31], including dysfunction in antioxidative response pathways [32,33,34], which further compromises the ability of cells to manage oxidative stress [35,36]. Unhealthy aging amplifies this decline, further weakening cellular oxidative stress resistance. Moreover, many of the risk factors for COPD, such as smoking, environmental pollutants [37], and chronic infections, further compromise these stress resilience mechanisms, thereby promoting unhealthy aging and increasing susceptibility to the development of COPD. Genetic predisposition also plays a role, as the antioxidant system’s efficiency and capacity can vary significantly based on the human genotype, influencing individual vulnerability to oxidative stress, ROS-mediated macromolecular damage, and associated diseases [38,39,40,41]. Additionally, impaired cellular stress resilience, including diminished antioxidant capabilities, amplifies the deleterious impact of ROS, perpetuating a cycle of oxidative damage and deteriorating tissue function [42,43,44,45,46,47,48,49,50,51]. This imbalance not only promotes the progression of chronic conditions like COPD, but also compromises the repair and adaptive capacity of cells and tissues, further accelerating the disease pathology.

In this context, trace elements like iron, copper, zinc, selenium, and magnesium are crucial for the antioxidant defense mechanisms in COPD [52]. These trace elements are essential cofactors for various enzymes and proteins involved in antioxidant defense mechanisms, inflammatory processes, oxidative stress regulation, and cellular repair mechanisms [53,54]. Iron, copper, and zinc, in particular, are key cofactors in regulating airway inflammation and oxidative pathways [55]. The imbalance of these elements can severely affect biological functions and contribute to COPD progression [56]. For example, zinc is vital for the protein structure, antioxidant defense, and immune regulation [57]. Copper is necessary for the function of many enzymes, while magnesium may aid in bronchodilation and improve lung function [58,59,60]. Trace elements play an important role in the management of both stable COPD and AECOPD. Their involvement in antioxidant defense, immune response regulation, and respiratory muscle function underscores their therapeutic potential. Monitoring and addressing deficiencies in iron, magnesium, zinc, calcium, selenium, copper, and manganese may help slow disease progression, reduce the frequency of exacerbations, and improve overall outcomes.

Trace elements can serve as both markers of oxidative stress and predictors of COPD exacerbations or progression. Several studies have linked low serum magnesium levels to the frequency and severity of exacerbations [61,62]. The regular monitoring of trace element levels and targeted supplementation could open new therapeutic opportunities for managing the disease [63,64]. Research indicates that variations in the serum levels of these trace elements are associated with exacerbations, disease severity, and overall outcomes [65,66]. Supplementation with trace elements, alongside standard treatment, may improve therapeutic outcomes, reduce disease progression, and facilitate bronchodilation [65,66]. Therefore, the role of antioxidants in COPD is significant not only in diagnosis, but also in treatment, warranting further research.

The purpose of this study is to review the role of trace elements in the pathogenesis of COPD, focusing on iron, copper, zinc, selenium, magnesium, manganese, and calcium. We will examine the impact of these elements on oxidative stress and lung function and present the latest research findings suggesting new diagnostic and therapeutic approaches. The studies presented in this review differ in their objectives, methodological approaches, and findings, but they share a common thread of identifying new directions for future research regarding the role of trace elements in heterogeneous COPD patient populations. The potential role of trace elements in predicting COPD exacerbations, their involvement in inflammatory and oxidative stress processes, their participation in disease pathophysiology, and their active role in treatment still require an extensive investigation. Thus, the role of trace elements is significant not only in the pathology of the disease, but also in the development of treatment strategies used in clinical practice. Regulating oxidative stress and maintaining trace element homeostasis holds promising potential for COPD treatment, although further research is necessary to fully understand and apply these clinical strategies.

## 2. Methods

A comprehensive literature search was conducted using PubMed, Web of Science, Cochrane Central Register of Controlled Trials (CENTRAL), and Google Scholar databases, covering studies from their inception to 1 October 2024. The primary objective was to identify randomized controlled trials (RCTs) and human clinical studies investigating the role of trace element supplementation in managing COPD.

Search strategies utilized specific Medical Subject Headings (MeSH) terms and keywords, including chronic obstructive pulmonary disease, exacerbation, spirometry, respiratory function, physical activity, quality of life, zinc, copper, iron, selenium, magnesium, manganese, calcium, trace element supplementation, and micronutrient supplementation. Duplicate entries were removed before screening titles and abstracts. Studies that did not meet the inclusion criteria were excluded. The remaining articles underwent full-text review. The selection process is illustrated in a flowchart (Figure 1).

The review adhered to the Population, Intervention, Comparison, and Outcomes (PICO) framework. It focused on the impact of trace element supplementation on respiratory function, physical activity, systemic inflammatory markers, and quality of life in COPD patients. A narrative synthesis approach was employed, integrating findings from cohort analyses, randomized controlled trials, cross-sectional studies, and clinical investigations.

Inclusion and exclusion criteria were established to maintain focus and relevance. It is important to note that the methodological quality of the included studies (e.g., randomization, blinding, and study rigor) was not explicitly used as a criterion for inclusion. This limitation should be considered when interpreting the findings of this review.

Inclusion criteria:Study population: patients aged 40 years and older with a confirmed diagnosis of COPD.Intervention: supplementation with trace elements, including zinc, copper, iron, selenium, magnesium, manganese, and calcium.Outcome measures: lung function (spirometry), physical activity (6 min walk test, incremental shuttle walk test), systemic inflammatory markers (C-reactive protein, interleukins, tumor necrosis factor-alpha), quality of life (COPD Assessment Test, St George’s Respiratory Questionnaire, EuroQol-5D), exacerbation frequency, and mortality risk.Study design: randomized controlled trials, human clinical trials, and follow-up cohort, retrospective, and cross-sectional studies.Published articles: articles indexed in PubMed, Web of Science, Cochrane Central Register of Controlled Trials (CENTRAL), and Google Scholar.

Exclusion criteria:Preclinical studies (in vitro studies and intervention studies involving animal models).In vitro studies.Interventions focusing on macronutrients.Dietary advice without intervention.

A total of 35 articles were included in this review, comprising data from 6751 COPD patients.

## 3. Results

### 3.1. Clinical Research and the Effect of Zinc Supplementation in COPD

Patients with COPD are often prone to a zinc deficiency, which may elevate their risk of infectious diseases [67,68]. Research suggests that zinc supplementation could help reduce exacerbations, improve respiratory function, and support overall patient health [63,69]. One study indicates that measuring the zinc concentration in hair might be a useful method for assessing a chronic zinc deficiency in these patients [70].

In a study by Kirkil G et al. [68], respiratory function tests and various biochemical parameters were evaluated in stable COPD patients and healthy controls. Patients received daily zinc picolinate supplementation (22 mg) for eight weeks. At both the start and end of the study, the levels of malondialdehyde (MDA), superoxide dismutase (SOD), catalase (CAT), and zinc were measured. After eight weeks, a significant increase was observed in both the SOD activity and serum zinc levels, though no significant changes were seen in the MDA, CAT levels, or respiratory function tests, including a forced expiratory volume in one second (FEV_1_) and the FEV_1_/FVC ratio. These findings suggest that zinc supplementation improved the oxidative–antioxidative balance without significantly impacting the respiratory function over the study period.

In an other study, Isbaniah F et al. [71] investigated whether Echinacea purpurea (EP) combined with ascorbic acid and micronutrients such as zinc, selenium, and vitamin C (EP+) could reduce COPD exacerbations triggered by upper respiratory tract infections. The EP+ treatment group experienced milder and shorter exacerbation episodes following an upper respiratory tract infection compared to the placebo group, with the treatment demonstrating a favorable safety profile. Gouzi F et al. [72] randomized sixty-four hospitalized COPD patients undergoing pulmonary rehabilitation into two groups: one receiving oral antioxidant supplementation tailored to address previously observed deficiencies (antioxidant group: α-tocopherol: 30 mg/day, ascorbate: 180 mg/day, zinc gluconate: 15 mg/day, selenomethionine: 50 μg/day) and a placebo group. The supplementation significantly increased the α-tocopherol/γ-tocopherol ratio, selenium levels (+58 ± 20%, *p* < 0.001, and +16 ± 5%, *p* < 0.01), muscle strength (+11 ± 3%, *p* < 0.001), and serum total protein (+7 ± 2%, *p* < 0.001). It also showed a trend toward increasing the proportion of type I muscle fibers (+32 ± 17%, *p* = 0.07), suggesting that antioxidant supplementation (including vitamins C and E, zinc, and selenium) may benefit COPD patients during pulmonary rehabilitation.

El-Attar M et al. [73] evaluated the effect of the intravenous administration of trace elements (selenium, manganese, and zinc) on the duration of the mechanical ventilation necessary in COPD patients. The intravenous supplementation group showed a significant reduction in the mechanical ventilation duration (9.4 ± 7.3 days) compared to the placebo group (17.8 ± 7.6 days, *p* = 0.013). This study highlights the role of oxidative stress and trace element levels in the need for mechanical ventilation among COPD patients.

Koç U. et al. [74] conducted a study to assess trace element levels during acute COPD exacerbations. They found a positive correlation between serum copper levels and both the erythrocyte sedimentation rate and leukocyte count (*p* < 0.05, r = 0.5; *p* < 0.05, r = 0.4). The serum copper and selenium levels were significantly elevated in stable COPD patients compared to healthy controls (*p* < 0.01). This finding suggests that higher copper and selenium levels in the stable phase may reflect chronic inflammation, while lower copper levels during acute exacerbations could indicate its utilization. Serum zinc levels were higher in healthy controls, potentially reflecting the combined effects of chronic inflammation and trace element consumption in COPD. These three elements are strongly interconnected in their roles within inflammatory pathways: elevated copper and selenium levels in stable COPD may signify ongoing chronic inflammation, while decreased copper levels and a zinc deficiency during acute exacerbations suggest the active involvement and utilization of these elements in inflammatory responses. All three trace elements play critical roles in antioxidant defense, which is essential for slowing COPD progression. These findings emphasize the need for further large-scale studies to better understand their therapeutic implications. Table 1 summarizes clinical studies investigating the effects of zinc in COPD.

### 3.2. Effects of Iron Therapy on Oxidative Stress and Quality of Life in COPD Patients

The study by Pérez-Peiró et al. [75] aimed to investigate whether iron therapy improves systemic oxidative stress in COPD patients with a non-anemic iron deficiency. Their results demonstrated that iron replacement significantly reduced oxidative stress markers and improved glutathione levels in stable, severe COPD patients.

Martín-Ontiyuelo et al. [76] assessed whether intravenous iron therapy enhances the exercise capacity, quality of life, and daily physical activity in COPD patients. Participants were randomized in a 2:1 ratio to receive intravenous ferric carboxymaltose or a placebo. The COPD Assessment Test (CAT) scores significantly decreased in the treatment group, indicating an improvement in their quality of life. During iron therapy, 52.3% of participants achieved the primary endpoint, suggesting that iron replacement improved the exercise capacity, compared to only 18.2% in the placebo group.

In a pilot study conducted by Grasmuk-Siegl et al. [77], patients with a non-anemic iron deficiency underwent a 6 min walk test and cardiopulmonary exercise testing. The study aimed to evaluate the effects of intravenous ferric carboxymaltose on the exercise capacity and quality of life in COPD patients. Their findings indicated that four weeks after therapy, the 6 min walking distance increased by an average of 34.7 m, and VO_2_max also improved, accompanied by a notable enhancement in their quality of life.

According to the study by Santer et al. [78], intravenous iron therapy may have a beneficial effect on the exercise capacity (6MWD) and quality of life of COPD patients, although improvements in oxygen levels are limited. The results indicated that iron therapy did not lead to significant changes in oxygen saturation, but it did improve the exercise capacity and reduce dyspnea, positively impacting the patients’ daily quality of life. However, the treatment did not result in significant improvements in other clinical parameters, and hypophosphatemia was more frequently observed as a potential side effect. Overall, while intravenous iron therapy appears promising for improving the exercise capacity and quality of life in COPD patients, its effects are not unequivocally favorable in all aspects. The findings emphasize the need for further research to confirm these results and investigate the long-term effects of iron therapy in COPD management. The study’s positive outcomes were primarily limited to secondary measures, highlighting the need for more trials with exercise-related variables as the primary outcomes. It was also a small, single-center study, which limits its generalizability. Additionally, pulmonary vascular effects were not fully explored, particularly in relation to pulmonary artery pressures, which could be important for identifying which COPD patients might benefit most from iron therapy, such as those with exercise-induced pulmonary hypertension or comorbid heart failure. Table 2 summarizes the clinical trials on iron therapy in COPD patients.

### 3.3. The Role of Magnesium Sulfate in COPD Exacerbation Management

Studies investigating the effects of magnesium sulfate in the exacerbation of chronic obstructive pulmonary disease indicate that it can significantly enhance lung function and exercise tolerance, particularly when combined with other bronchodilators [79,80]. Magnesium sulfate exerts its effects in COPD exacerbations through multiple mechanisms. It acts as a calcium channel blocker, reducing smooth muscle contraction, and inhibits the acetylcholine release, leading to bronchodilation. Additionally, it has anti-inflammatory properties, decreasing the neutrophil activity in the airways and the histamine release from mast cells. Intravenous magnesium sulfate improves the lung function and respiratory rate, especially when combined with other bronchodilators. For instance, Mukerji et al. [80] found that the intravenous administration of magnesium sulfate during COPD exacerbations improved lung function metrics such as FEV_1_ and FVC alongside standard bronchodilator therapy. Similarly, Amaral et al. [81] reported that magnesium infusion resulted in significant improvements in the maximal exercise capacity and respiratory mechanics, including a reduction in static lung hyperinflation.

In a study by Skorodin et al. [82], magnesium sulfate was shown to be both safe and effective in managing COPD exacerbations, offering a greater bronchodilator effect compared to beta-agonist treatment alone. Furthermore, a double-blind randomized clinical trial conducted by Vafadar Moradi [83] demonstrated that intravenous magnesium sulfate infusion significantly improved the Peak Expiratory Flow Rate (PEFR), lung function, and the Dyspnea Severity Scale (DSS) compared to the placebo, with no side-effects reported and no patients requiring intubation.

However, it is important to note that magnesium sulfate alone did not consistently yield significant bronchodilatory effects. For instance, in the study by Abreu González et al. [84], magnesium sulfate did not produce a marked bronchodilatory effect on its own, but did enhance the dilation induced by beta-agonists, showing greater increases in FEV_1_ compared to the placebo group. Similarly, research by Solooki et al. [85] found that PEFR and FEV_1_ measurements in the group receiving magnesium sulfate did not show a significant improvement over the placebo at either 45 min or 3 days post-treatment. Collectively, these studies suggest that while magnesium sulfate can improve the lung function and exercise tolerance in COPD exacerbations, particularly in combination with other therapies, its standalone effectiveness may vary. Further research is needed to fully understand its role and optimize treatment protocols (Table 3).

### 3.4. The Role of Selenium in Lung Function and COPD

Selenium is an essential trace element recognized for its anti-inflammatory and antioxidant properties [88]. Despite the known benefits of selenium, evidence regarding its direct relationship with COPD remains limited. A cross-sectional study utilizing data from the National Health and Nutrition Examination Survey (NHANES) found a negative correlation between the selenium intake and the risk of developing COPD. Specifically, the adjusted odds ratio (OR) for the disease in individuals with the highest selenium intake (>122.0 μg/day) was 0.80 (95% CI: 0.71–0.91, *p* < 0.001) [89].

Antioxidants play a critical role in reducing oxidative damage to the lungs, providing protection against chronic respiratory diseases and slowing the decline in lung function. El-Attar et al. [73] assessed the intravenous administration of trace elements, including selenium, manganese, and zinc, and its effects on the duration of mechanical ventilation in COPD patients. Serum levels of these trace elements were measured by using inductively coupled plasma spectroscopy. Their findings indicated that trace element supplementation significantly reduced the duration of mechanical ventilation to 9.4 ± 7.3 days compared to 17.8 ± 7.6 days in the placebo group (*p* = 0.013), suggesting that the trace element support may shorten the time required for mechanical ventilation.

In a randomized trial by Gouzi et al. [72], 64 COPD patients received antioxidant supplementation over 28 days (α-tocopherol: 30 mg/day, ascorbic acid: 180 mg/day, zinc gluconate: 15 mg/day, selenomethionine: 50 μg/day) during a pulmonary rehabilitation program that included 24 sessions of moderate-intensity exercise. This study measured outcomes related to muscle endurance, oxidative stress, and rehabilitation efficacy. The results showed an increased α-tocopherol/γ-tocopherol ratio and serum selenium levels (+58 ± 20%, *p* < 0.001; +16 ± 5%, *p* < 0.01, respectively), along with improved muscle strength (+11 ± 3%, *p* < 0.001) and serum total protein levels (+7 ± 2%, *p* < 0.001). The prevalence of muscle weakness in the antioxidant supplementation group decreased from 30% to 10.7% (*p* < 0.05), suggesting that antioxidant supplementation could be a beneficial therapeutic adjunct during pulmonary rehabilitation.

Another study by Santos et al. [90] found that plasma selenium levels were significantly lower in COPD patients compared to the controls (ex-smokers: 0.030 ± 0.019 mg/L, smokers: 0.032 ± 0.024 mg/L, controls: 0.058 ± 0.023 mg/L; *p* < 0.01), particularly among those with lower arterial oxygen pressure. This finding implies that oxidative damage and electrolyte imbalance are more closely related to chronic lung disease rather than being direct effects of smoking. Furthermore, an analysis using data from the Third National Health and Nutrition Examination Survey (NHANES III) investigated the relationship between lung function and the antioxidant nutrient intake [91]. This study found that both dietary and serum antioxidant levels were positively associated with lung function, assessed via the FEV_1_-to-height ratio, indicating that most antioxidants demonstrated an independent positive effect on the respiratory performance. Research suggests that selenium’s antioxidant and anti-inflammatory properties are beneficial in lung health, with evidence showing its potential to lower COPD risk, decrease the need for extended mechanical ventilation, enhance muscle strength, and support an improved respiratory function. The relationship between antioxidants and lung function is based on the role of antioxidants in neutralizing free radicals, which cause oxidative stress in lung tissues. Antioxidants, such as selenium, can help protect the lungs from inflammation and damage. This relationship may be stronger in more severe diseases, as chronic lung conditions, such as COPD, lead to increased oxidative stress.

### 3.5. Manganese and COPD: Potential Role in Disease Pathogenesis

Manganese (Mn) is an essential trace element that plays a crucial role in various physiological processes, including antioxidant defense, carbohydrate and lipid metabolism, and the neutralization of free radicals. Manganese-superoxide dismutase (Mn-SOD), a key enzyme located in the mitochondria, provides primary protection against oxidative stress, particularly from ROS [73]. Given that oxidative stress is central to the pathogenesis of COPD, manganese’s role in this condition has garnered increasing attention.

A deficiency of manganese may compromise the lung’s defense mechanisms, particularly by impairing the mitochondrial function, which can exacerbate ROS-induced cellular damage. However, the levels of manganese and their effects in COPD patients remain incompletely understood, as research presents mixed findings [92,93,94,95]. Some studies have reported decreased manganese levels in affected individuals, while others have found no significant differences compared to healthy subjects [92,93,94,95]. This inconsistency raises questions about whether manganese’s impact may vary depending on the disease stage or individual factors such as diet and environmental exposure (e.g., smoking, air pollution).

Manganese supplementation could represent a potential therapeutic approach, particularly for individuals whose disease progression may be linked to manganese deficiency [96]. Enhancing manganese’s antioxidant properties could potentially improve respiratory function and reduce oxidative damage to lung tissues caused by ROS [97]. The effects of manganese and other antioxidants on COPD can vary in different ways. Manganese and other trace elements, such as selenium, may have synergistic effects that can improve lung function and reduce oxidative stress [98]. Antioxidant minerals like selenium, copper, manganese, and zinc are also essential, as they are required for the proper function of antioxidant enzymes [69,99]. For example, selenium is necessary for the activity of glutathione peroxidases, which help neutralize hydrogen peroxide. Zinc acts as a cofactor for antioxidant enzymes, while manganese plays an important role as a key component of Mn-superoxide dismutase, which protects against mitochondrial oxidative stress [97]. According to research, the impact of these elements may depend on the disease stage, individual diet, and environmental factors [100]. When using multiple trace elements together, it is important to maintain a balance to avoid potential negative interactions. However, because aan excessive manganese intake has known toxic effects, such as neurotoxicity, careful planning is essential when considering supplementation to avoid adverse outcomes [73,101].

Further investigation into the role of manganese in chronic respiratory diseases is necessary to clarify this trace element’s impact on disease onset and progression. Additional studies are needed to determine whether supplementation could benefit affected patients, particularly in establishing an optimal, safe dosage. The therapeutic potential of manganese may be most relevant for individuals in whom oxidative stress and inflammatory responses are significant factors exacerbating symptoms.

### 3.6. The Role of Copper in COPD: Biomarker Potential and Therapeutic Implications

Copper is an essential trace element involved in numerous biological processes, including the function of antioxidant enzymes and the regulation of inflammatory responses. It serves as a cofactor for several enzymes, such as superoxide dismutase, which plays a crucial role in neutralizing ROS [59,102]. However, excessive copper levels can lead to pro-oxidant effects, potentially exacerbating oxidative damage in the airways and contributing to the progression of COPD.

Research indicates that serum copper levels may be elevated in patients with COPD, correlating with disease severity and an increased risk of exacerbations [59,102]. This suggests that copper could serve as a potential biomarker for both the diagnosis and monitoring of COPD. Moreover, regulating copper homeostasis presents a promising therapeutic target, particularly for patients where oxidative stress is a significant factor.

A study by Kunutsor et al. [59] investigated the association between serum copper/zinc (Cu/Zn) ratios and COPD risk. Their multivariate analysis revealed that a one-unit increase in the Cu/Zn ratio significantly raised the risk of COPD (hazard ratio, HR: 1.81; 95% CI: 1.08–3.05). The adjusted HR for copper alone was found to be 3.17 (95% CI: 1.40–7.15), indicating that while copper is vital for maintaining antioxidant defenses, excessive levels may contribute to the pathogenesis of COPD. The study concluded that higher serum Cu/Zn ratios and copper concentrations were linearly associated with an increased COPD risk.

Similarly, research by Fei et al. [103] supports the notion that elevated copper levels may heighten the risk of chronic respiratory conditions. Isik et al. [104] found that serum copper and malondialdehyde levels were significantly higher in COPD patients compared to controls, highlighting a potential link between copper and oxidative stress. Conversely, a preliminary study by Valkova et al. [105] suggested that elevated copper and zinc levels might not always be harmful and could even serve as prognostic biomarkers for disease progression.

Additionally, studies have shown that exacerbations of COPD are associated with increased levels of copper, zinc, and lipid peroxidation (MDA). Tanrikulu et al. [66] demonstrated that oxidative stress intensifies during these exacerbations, evidenced by decreased coenzyme Q10 levels, shifts in the Cu/Zn ratio, and elevated copper and zinc concentrations—reflecting the body’s defense mechanisms and ongoing inflammation. The elevated levels of copper and zinc in COPD are likely a consequence of oxidative stress and inflammatory responses associated with COPD, which may increase during disease progression and exacerbations. Thus, copper and zinc levels in COPD not only reflect the body’s response to oxidative stress, but also indicate the progression and exacerbation of the disease. Therefore, regulating copper levels is essential because, while they can provide antioxidant effects when balanced, excessive levels may lead to harmful outcomes. Future research should aim to deepen the understanding of copper’s role in the pathogenesis of COPD and explore how regulating its levels could improve patient outcomes and inform therapeutic strategies.

### 3.7. Calcium and COPD: Its Role in Disease Pathogenesis

Calcium (Ca) is crucial for a wide range of cellular functions, such as muscle contraction, neural signaling, and cell growth and differentiation [106]. It plays an essential role in regulating processes like inflammatory responses, smooth muscle cell contraction, and apoptosis, all of which are key factors in the pathogenesis and progression of COPD [107]. In patients with COPD, elevated intracellular calcium levels may exacerbate lung inflammation by promoting the release of inflammatory mediators, which further impair the lung function and worsen disease outcomes [107]. Moreover, calcium is involved in airway smooth muscle contraction, contributing to airway narrowing and bronchial hyper-reactivity [108]. It is well known that the release of these mediators is a process regulated by intracellular calcium.

Calcium dysregulation is also linked to bone metabolism, with many COPD patients experiencing bone loss due to long-term corticosteroid use and reduced physical activity [109]. Studies have shown that alveolar macrophages in COPD patients exhibit impaired bacterial phagocytosis and altered cytokine secretion—processes that are calcium-dependent [110,111,112]. Provost et al. [113] investigated the influence of calcium on macrophage responses to nontypeable Haemophilus influenzae, hypothesizing that increasing extracellular calcium levels would restore phagocytosis and cytokine secretion in monocyte-derived macrophages. Their findings revealed that extracellular calcium significantly enhanced phagocytosis and cytokine secretion (IL-8, TNF-alpha, and macrophage inflammatory protein [MIP]-1 alpha and -1 beta), as well as the expression of bacterial recognition receptors such as CD16 and the Macrophage Receptor with a Collagenous Structure (MARCO). These results suggest the therapeutic potential of calcium in improving macrophage function, particularly in reducing exacerbations and chronic bacterial infections.

A case-control study by Hirayama et al. [114] examined the relationship between the dietary mineral intake and the risk of developing chronic respiratory conditions. Their analysis indicated that a higher calcium intake was significantly associated with a reduced likelihood of developing COPD. Similarly, an increased iron intake showed an inverse relationship with COPD risk.

Research by Andreotta et al. explored the effects of inhaled salts on smoke-induced inflammation, specifically investigating whether cations could modify airway surface liqui properties to mitigate inflammation [115]. Their results indicated that inhaled calcium salts moderated cellular inflammation and reduced elevated chemokine and cytokine levels associated with smoke exposure, unlike sodium and magnesium salts. This suggests that inhaled calcium salts may help reduce lung inflammation at both biological and physical levels. Another study [116] assessed serum sodium, potassium, and ionized calcium levels during exacerbations, finding significantly lower levels in affected individuals compared to controls, with the lowest values occurring during severe episodes.

Chronic respiratory diseases are often associated with malnutrition, which can worsen clinical outcomes. Proper nutrition, including an adequate calcium intake, may substantially improve the clinical status and quality of life in affected individuals. One study found that 52% of COPD patients had an energy intake below the recommended levels, and over 75% did not meet the recommended daily intake (RDI) for several essential nutrients, including calcium, potassium, folic acid, vitamin D, and vitamin A [117]. These deficiencies were more pronounced than those observed in age-matched individuals with healthy respiratory systems. A vitamin D deficiency is also common among COPD patients, with the severity increasing in advanced disease stages. While vitamin D supplementation plays a well-established role in reducing the risk of falls and osteoporosis-related fractures, there is limited strong evidence to suggest it slows the decline in lung function [118].

The multifaceted role of calcium in inflammation, airway smooth muscle function, and bone metabolism presents targeted therapeutic opportunities in managing chronic respiratory conditions. Further research is needed to understand how optimizing the calcium intake and regulation might improve outcomes for patients suffering from these diseases.

## 4. Discussion

This review article comprehensively evaluates the impact of several essential minerals (zinc, iron, magnesium, selenium, manganese, copper, and calcium) on the pathogenesis of COPD. These trace elements are not only critical to disease progression, but are also essential for managing symptoms in affected individuals. COPD is a progressive airway disorder characterized by airflow limitation, chronic inflammation, and episodes of acute exacerbations. AECOPD involve a sudden worsening of respiratory symptoms, often triggered by infections, pollutants, or other environmental factors, and frequently require hospitalization or urgent care. Moreover, stable COPD is associated with a more predictable pattern of symptoms and less intense inflammation; however, oxidative stress and tissue remodeling persist. Trace elements play a crucial role in both stable COPD and during exacerbations as they are fundamental to enzymatic functions, the redox balance, and immune regulation. Their involvement in antioxidant defense and respiratory muscle function underscores their therapeutic potential. Monitoring and addressing deficiencies in iron, magnesium, zinc, calcium, selenium, copper, and manganese may help slow disease progression, reduce the exacerbation frequency, and improve overall outcomes. Trace element deficiencies exacerbate the COPD pathophysiology through several mechanisms. The reduced activity of antioxidant enzymes leads to increased oxidative stress. Low levels of zinc, copper, or magnesium impair the immune function, increasing susceptibility to infections, which are major triggers of AECOPD. Hypoxia may worsen due to disruptions in oxygen transport and utilization, often linked to iron or magnesium deficiencies. Furthermore, airway inflammation intensifies and the respiratory muscle performance declines, exacerbating the clinical manifestations of COPD. As summarized in Table 4, addressing these deficiencies offers significant potential to mitigate these effects and improve patient outcomes.

As COPD progresses, a number of extrapulmonary complications, such as sarcopenia, often emerge, characterized by a reduction in muscle mass and strength [119]. Factors such as systemic inflammation, hypoxia, oxidative stress, disruptions in protein metabolism, and mitochondrial dysfunction collectively impair muscle function, significantly affecting the patient’s quality of life, increasing hospital stay durations, and raising the mortality risk [100]. Several minerals have been shown to exert indirect beneficial effects on muscle function and are associated with muscle mass. Recent research indicated that patients with severe COPD and an iron deficiency exhibit low serum ferritin levels, which correlate significantly with their walking distance in the six-minute walk test, a marker of muscle function decline [120]. Calcium is essential for maintaining proper muscle and nerve function, while selenium exhibits antioxidant properties. Observational studies have demonstrated that the serum calcium [121] and selenium intake [122] are significantly associated with a reduced muscle mass. Selenium supplementation may enhance the calcium release and improve the skeletal muscle performance [123]. Antioxidant supplements containing zinc gluconate and selenium, combined with pulmonary rehabilitation, improve the quadriceps strength and increase the proportion of type I muscle fibers in COPD patients [72]. Magnesium plays a critical role in predicting disease severity as it can reduce systemic inflammation and pro-inflammatory cytokine levels [124]. In COPD patients, ionized magnesium concentrations were significantly lower compared to healthy non-smokers, and the Ca/Mg ratio in both plasma and polymorphonuclear cells was significantly higher [125]. A randomized controlled trial demonstrated that the consumption of whey drinks enriched with magnesium and vitamin C led to an increased lean body mass, FFMI, and muscle strength in COPD patients [126]. Furthermore, a higher magnesium intake was significantly and positively correlated with an appendicular lean body mass [127], and the serum magnesium concentration was independently correlated with muscle strength [128]. Overall, mineral supplementation presents an effective therapeutic strategy for preserving muscle mass and strength in patients with COPD and sarcopenia. However, further research is required to develop optimal mineral supplementation protocols for these patients.

In the pathogenesis of COPD, the imbalance between oxidative stress and antioxidant defenses plays a crucial role, with zinc emerging as a promising supplemental therapeutic element [129,130,131]. Through its antioxidant properties, zinc contributes to the function of SOD, which is essential for neutralizing reactive oxygen species and mitigating oxidative damage in the lungs [69,132]. Additionally, zinc plays a pivotal role in supporting immune function, as a zinc deficiency can lead to impaired immunity, increasing the risk of respiratory infections and disease exacerbations [133,134]. Studies indicate that COPD patients with low zinc levels are more prone to inflammation and tissue damage, which further worsens the respiratory function and contributes to disease progression [58,69,135]. Zinc supplementation in COPD may offer various benefits, including an enhanced antioxidant capacity, reduced inflammation, and strengthened immune functions, potentially lowering the frequency of infections and disease exacerbations [98]. Clinical trials have shown that dietary and oral zinc supplements can improve the lung function and general physical performance in patients, suggesting that zinc supplementation, integrated into standard COPD treatment protocols, could improve the patient’s quality of life [136,137,138]. Additionally, exploring the interactions between zinc and other trace elements, such as selenium and magnesium, could provide insights into their combined potential to enhance antioxidant defenses and reduce inflammation, thus supporting a slower progression of the disease. Various biomarkers are used to determine the zinc status, though none are perfect, and each has its limitations. Common biomarkers include the plasma or serum zinc concentration, urine zinc levels, zinc transporter 2 levels, and the activity of zinc-dependent metalloenzymes like SOD. However, factors such as inflammation, infections, the hydration status, or kidney function can influence these markers, making them insufficient on their own to accurately reflect the long-term zinc status. Therefore, a combination of methods is often required for a more precise assessment of a zinc deficiency or excess.

Iron is essential for all living organisms due to its ability to transition between oxidation states, playing a crucial role in various physiological processes, such as energy metabolism, immune function, oxygen transport, and neurotransmitter synthesis [64,139]. Recent studies highlight that an iron deficiency, especially a non-anemic iron deficiency, may exacerbate COPD severity [76,140,141]. Research by Pérez-Peiró et al. has demonstrated that iron supplementation reduces oxidative stress markers and enhances glutathione levels in stable, severe COPD patients, potentially improving their quality of life [75]. Preventing exacerbations—a key element in COPD management—significantly affects patients’ quality of life and survival [142,143,144]. Alongside appropriate bronchodilator therapy and inhaled corticosteroids, iron–polymaltose complexes, known to reduce malondialdehyde levels, may be beneficial [145]. A non-anemic iron deficiency is common in COPD patients and is associated with reduced exercise tolerance, which directly impacts their quality of life [146]. From a mechanistic standpoint, iron supplementation in COPD patients may reduce systemic inflammation and help restore skeletal muscle function [75]. Inflammatory processes elevate hepcidin levels, inhibiting iron absorption and mobilization, thereby contributing to an iron deficiency. Though limited, current clinical trials suggest that intravenous iron supplementation can enhance exercise tolerance in COPD patients [77,147]. However, side-effects (e.g., hypophosphatemia), treatment costs, and the need for repeated intravenous access limit iron supplementation as a standard treatment in COPD. Additional large-scale studies are needed to further understand its benefits and safety profile.

COPD exacerbations, defined as sudden flare-ups of the disease, are episodes characterized by the worsening of symptoms in patients diagnosed with COPD [2,148,149]. These exacerbations significantly impact patients’ quality of life and the progression of the disease [150,151]. Magnesium sulfate, known for its bronchodilatory effects, can be administered intravenously or via inhalation using a nebulizer to help alleviate respiratory issues [152]. Several studies suggest that magnesium sulfate can be effectively used as an adjunct therapy in the management of COPD exacerbations [63,79,80,82,84,153,154,155]. The role of magnesium during COPD exacerbations is becoming increasingly important, as a magnesium deficiency may exacerbate inflammatory responses [156]. As a natural muscle relaxant, magnesium plays a key role in regulating the tone of smooth muscles in the airways. Supplementation with magnesium may help reduce the frequency of COPD exacerbations by decreasing inflammation and improving the function of respiratory muscles [154]. The role of magnesium in exacerbations is a promising area that requires further research to optimize therapeutic options and disease management strategies. One study compared the inhalation and infusion of magnesium sulfate with ipratropium bromide, but did not identify significant differences between the effects of these treatments [157]. Magnesium sulfate may offer potential benefits as an adjunct therapy for acute COPD exacerbations, particularly considering that low serum magnesium levels are associated with an increased risk of exacerbations [158]. Intravenous magnesium sulfate, when used alongside bronchodilators, may reduce hospital admissions and improve lung function, especially during acute asthma exacerbations [159]. In contrast, there are no confirmed clinical benefits for inhaled magnesium sulfate, indicating that further research is needed to establish its usefulness.

Selenium, as an essential trace element, has garnered significant attention in the treatment of COPD due to its antioxidant properties that mitigate inflammatory processes [160]. A selenium deficiency is common among COPD patients and is associated with accelerated disease progression. Research, including the study by El-Attar et al. [73], demonstrates that the infusion of selenium, manganese, and zinc can significantly reduce the duration of mechanical ventilation (from an average of 17.8 days in the placebo group to 9.4 days in the treatment group, *p* = 0.013). This research highlights the beneficial impact of trace element supplementation in the management of severe COPD. Selenium plays a fundamental role in the synthesis of glutathione peroxidase, an enzyme with potent antioxidant and anti-inflammatory properties, effectively suppressing systemic responses to oxidative stress [161]. COPD patients exhibit lower serum selenium levels compared to healthy controls, and selenium supplementation has been shown to alleviate their symptoms [63,89]. The reduction in oxidative stress is achieved through the enhancement of glutathione peroxidase 1 (GPX1) activity, which is crucial for protecting the respiratory system. Low selenium levels have been linked to more frequent exacerbations and poorer clinical outcomes [162]. The oxidative stress that can lead to muscle dysfunction and atrophy in COPD arises from the overproduction of ROS and a lack of antioxidant defense [163,164]. Antioxidants, including selenium, improve muscle endurance and can reduce atrophy while also mitigating systemic inflammation [62,63]. Due to the severe antioxidant deficiencies in COPD patients, exercise alone may be insufficient to reduce oxidative damage; thus, the supplemental use of selenium and other antioxidants may be beneficial in enhancing exercise responses and in comprehensive pulmonary rehabilitation therapy [165,166]. Furthermore, selenium’s therapeutic effects occur through selenoproteins, which perform antioxidant and enzymatic functions. At least 25 selenoproteins are known in human tissues, each with distinct biological roles [167]. For example, glutathione peroxidases [168] and thioredoxin reductases [169] help cells defend against harmful oxidative effects. Selenium also influences the activity of thyroid hormones, which play an important role in the functioning of the endocrine system. In conjunction with vitamin E, selenium protects cell membranes and organelles from peroxidative damage, which is crucial for maintaining cellular health [170]. Additionally, selenium has been shown to modulate the immune response by regulating pro-inflammatory cytokines and improving the function of immune cells, which could potentially reduce the systemic inflammation seen in COPD [170,171]. Overall, selenium may play a promising role in the management of COPD, but further research is needed to develop effective therapeutic strategies.

Manganese, as an essential trace element, is receiving increasing attention in the management of COPD, primarily due to its antioxidant and anti-inflammatory properties [94]. Manganese plays a crucial role in the synthesis of SOD, an enzyme involved in protection against oxidative stress, thereby contributing to the reduction in respiratory inflammation [172]. A manganese deficiency is common among COPD patients and has been associated with disease exacerbation and increased inflammatory processes. Research indicates that manganese supplementation can improve the respiratory function and reduce the levels of inflammatory markers in lung tissues [173,174,175]. Additionally, manganese’s antioxidant effects may help protect the airways from oxidative stress, which plays a key role in the pathogenesis of COPD [13]. Studies suggest that manganese supplementation may contribute to a decrease in the frequency of COPD exacerbations and improve the patients’ overall quality of life [100,176]. While promising research supports the potential benefits of manganese, further investigations are needed to fully understand the precise mechanisms of action and to optimize the clinical application of manganese supplements in COPD management. Exploring the connections between a manganese deficiency and COPD may help develop more effective therapeutic strategies for this condition.

Copper is an essential trace element for the human body, participating in numerous enzymatic reactions and biochemical processes [103]. Notably, it serves as a key cofactor for antioxidant enzymes, such as SOD, which plays a central role in reducing oxidative stress. Additionally, copper is crucial in regulating inflammatory responses and supporting immune system functions, which is especially important in controlling the chronic inflammation often observed in COPD [103,177]. Maintaining adequate copper levels contributes to reducing oxidative stress, controlling inflammation, and supporting lung function [130]. Copper also plays a significant role in the cross-linking of elastin, a protein that maintains the elasticity of lung connective tissue. In cases of a copper deficiency, elastin cross-linking becomes impaired, weakening the connective tissue and potentially leading to structural changes in the lungs associated with emphysema-like damage. In a previous study, it was demonstrated that among non-smokers, higher copper concentrations in tap water were significantly associated with improved lung function values, including both FVC (*p* = 0.014) and FEV_1_ (*p* = 0.027), while no similar association was found among former or current smokers [178]. These findings suggest that the copper intake may be an important determinant of lung function, particularly in the non-smoking population, and further research is warranted to confirm these observations. Further clinical research is necessary to deepen our understanding of copper’s role in COPD. Future studies should focus on evaluating the use of copper and other antioxidant trace elements as adjunctive therapies for alleviating COPD symptoms, especially in terms of reducing inflammation and oxidative stress. Additionally, investigating the potential effects of copper supplementation on lung function improvement and exacerbation reduction could provide valuable insights into COPD management.

Adequate calcium levels play a crucial role in the management of COPD, particularly in maintaining muscle strength, supporting the respiratory muscle function, mitigating inflammatory processes, and preserving bone health [179]. Calcium’s role in supporting muscle function is of special importance in COPD, as the disease progression often leads to respiratory muscle weakness, which significantly contributes to worsening dyspnea, reducing the lung capacity, and limiting the physical performance in patients. Furthermore, calcium, in synergy with vitamin D, supports bone density, which is essential as COPD patients are at an increased risk of osteoporosis, especially with steroid-based therapies [180]. The supplementation of calcium and vitamin D has shown to reduce fracture risk due to osteoporosis and may improve lung function [181]. Epidemiological studies indicate an association between serum calcium levels and all-cause mortality [182,183]. Data from the UK Biobank and NHANES reveal a U-shaped relationship between the serum calcium concentration and mortality, suggesting that both excessively high and low calcium levels may elevate the mortality risk [182]. Additionally, population-based cohort studies have demonstrated a link between serum calcium levels and cardiovascular and cancer-specific mortality [182,183]. Mechanistic studies also suggest that intracellular calcium dysregulation in COPD patients may lead to pulmonary hypertension, which in severe cases can be fatal [106,107]. Further research is needed to clarify the precise effects of serum calcium levels on COPD prognosis and treatment outcomes. Some studies suggest that an adequate calcium intake may improve the immune response and reduce the risk of COPD exacerbations and bacterial infections, while inhaled calcium salts may help alleviate lung inflammation [58,179]. Optimizing calcium regulation presents a promising therapeutic approach in COPD management; therefore, ensuring an adequate calcium intake and monitoring serum levels are essential in the comprehensive treatment of COPD, as this approach may not only relieve respiratory symptoms, but also improve patients’ quality of life.

The following table (Table 5) summarizes the recommended daily intake (NRV—Nutrient Reference Value) of trace elements for patients with COPD, along with their known effects on the disease’s symptoms and progression; however, it does not constitute an official recommendation for a COPD-specific nutrient intake. Further research is required to validate the precise dosages, as they are influenced by numerous factors, including individual patient needs, the disease stage, comorbidities, and the presence of nutrient deficiencies.

Recommendations for future research

To advance therapeutic strategies in COPD management, focusing on molecular targets and mechanisms that can improve patients’ quality of life and slow disease progression, the following areas warrant further investigation:The effects of trace element supplementation and combination therapies: exploring the potential benefits of combined supplementation (e.g., zinc and selenium, or iron and selenium) to maximize the therapeutic efficacy.The optimization of trace element dosages: determining the optimal dosages of trace elements tailored to individual needs and disease stages to enhance treatment outcomes.The reliability of biomarkers: evaluating the accuracy and reliability of various biomarkers for identifying trace element deficiencies and monitoring the efficacy of supplementation therapies.

This review article comprehensively examines the roles of essential minerals—zinc, iron, magnesium, selenium, manganese, copper, and calcium—in the pathogenesis of COPD. These minerals are shown to be fundamental not only in disease progression, but also in symptom management. Zinc, with its antioxidant and immune-supporting properties, can reduce respiratory inflammation and prevent exacerbations. Iron’s antioxidative properties and respiratory support enhance the quality of life, while magnesium acts as an effective adjunct in respiratory muscle function enhancement. Selenium and manganese contribute to respiratory inflammation reduction and tissue protection as antioxidants, and copper, crucial for antioxidant enzymes, may serve as a biomarker for disease progression. Calcium is essential for muscle strength and bone health, especially in COPD patients on long-term corticosteroid therapy. This review emphasizes that further research is necessary to clarify these trace elements’ specific roles and therapeutic potentials in COPD, as these minerals could improve the respiratory function, alleviate symptoms, and enhance the quality of life for patients.

## 5. Limitations

This review is non-systematic; therefore, certain findings or research areas may have been omitted from the analysis. The examination of the relationship between trace elements and COPD is complex and multifaceted, and not all studies meet the same quality standards, which complicates drawing definitive conclusions. Additionally, the methodological differences across studies, the varying techniques used to measure trace elements, and the different stages of COPD investigated contribute to the heterogeneity of the results.

A significant limitation of this review is that the scientific quality of the included studies was not an inclusion or exclusion criterion. While the narrative review encompasses a wide range of studies, including randomized controlled trials and observational studies, the variability in the methodological rigor among the included studies may affect the reliability and generalizability of the conclusions. Furthermore, a portion of the findings is based on cross-sectional studies, which do not allow for the clear determination of causal relationships. In particular, the presentation of the role of selenium, manganese, copper, and calcium highlights a critical limitation: the lack of sufficient evidence and the low sample sizes of the included studies.

## 6. Conclusions

This review aims to thoroughly investigate and map the role and effects of various trace elements—including zinc, iron, magnesium, selenium, manganese, copper, and calcium—in the pathogenesis, progression, and symptom management of COPD. The adequate intake of these minerals may be of paramount importance in COPD treatment, especially in reducing exacerbations and enhancing the patient’s quality of life. Future research should focus on optimizing the intake of critical minerals, particularly iron, magnesium, selenium, and calcium, to support improved patient outcomes. Developing effective therapeutic protocols will require a multidisciplinary approach, emphasizing close collaboration among physicians, dietitians, and other healthcare professionals. A clear understanding of how these minerals impact the daily quality of life and symptom management is essential to successful COPD treatment. Furthermore, future studies must consider patients’ subjective experiences and feedback to ensure that therapeutic approaches are responsive to their specific needs. In summary, further research is needed to deepen our understanding of the mechanisms and clinical applications of these elements in COPD management to meaningfully enhance the quality of life for patients living with this condition.

## Figures and Tables

**Figure 1 nutrients-16-04118-f001:**
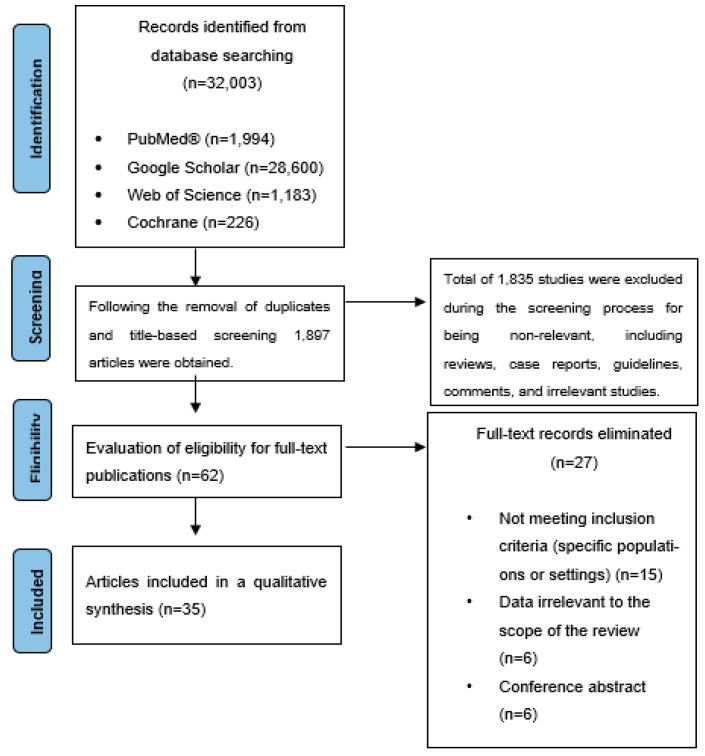
Flowchart illustrating the selection process of the included articles.

**Table 1 nutrients-16-04118-t001:** Summary of clinical studies investigating the effects of zinc in COPD.

Study	Design	Mean Follow-Up	Country	Sample Size	Average Age (Year)	Sex Male/Female	Intervention	Main Results
Kirkil G et al. [68]	RCT	8 weeks	Turkey	45	61.0 ± 2.8	100% male	22 mg zinc picolinate/day, 8 weeks	Increased SOD (*p* = 0.029) and zinc (*p* < 0.001); no significant changes in MDA, CAT, FEV_1_, or FEV_1_/FVC
Isbaniah F et al. [71]	RCT	4 weeks	Indonesia	108	65.8	97%/3%	Ciprofloxacin for 7 days (acute URTI); daily EP, EP+ (EP with zinc, selenium, vitamin C), or placebo	EP+ group had milder, shorter exacerbations; EP alone showed no significant benefit
Gouzi F et al. [72]	RCT	28 days	France	64	62.4 ± 6.5	50%/50%	Oral antioxidants: α-tocopherol (30 mg/day), ascorbate (180 mg/day), zinc gluconate (15 mg/day), and selenomethionine (50 μg/day)	Muscle weakness reduced in PR group (30% to 10.7%, *p* < 0.05); type I fibers increased (+32 ± 17%, *p* = 0.07)
El-Attar M et al. [73]	RCT	-	Egypt	120	62.8 ± 9.0	97%/3%	Daily IV: sodium selenite (100 µg), zinc chloride (2 mg), manganese (0.4 mg)	Mechanical ventilation duration decreased (9.4 ± 7.3 days vs. 17.8 ± 7.6 days, *p* = 0.013)
KOÇ U et al. [74]	CS	8 weeks	Turkey	30	68.8 ± 9.5	60%/40%	No intervention	Serum zinc levels were lower in stable COPD patients compared to healthy controls

Abbreviations: COPD: chronic obstructive pulmonary disease; IV: intravenous; RCT: randomized controlled trial; CS: cross-sectional study; SOD: superoxide dismutase; MDA: malondialdehyde; CAT: catalase; FEV_1_: forced expiratory volume in 1 s; FVC: forced vital capacity; URTI: upper respiratory tract infection; EP: echinacea purpurea; PR: pulmonary rehabilitation; μg: microgram; mg: milligram; %: percent.

**Table 2 nutrients-16-04118-t002:** Summary of clinical trials on iron therapy in COPD patients.

Study	Design	Mean Follow-Up	Country	Sample Size	Average Age (Year)	Sex Male/Female	Intervention	Main Results
Pérez-Peiró M et al. [75]	RCT	4 weeks	Spain	62	66.8 (7.3)	73%/27%	Intravenous ferric carboxymaltose (Ferinject^®^ Ferinject^®^, Vifor, St. Gallen, Switzerland) or placebo	MDA-protein adducts and 3-nitrotyrosine decreased; glutathione (GSH) increased. Hepcidin levels correlated with ferritin
Martín-Ontiyuelo C et al. [76]	RCT	4 weeks	Spain	66	68.0 (62–72)	64%/36%	Single dose of intravenous ferric carboxymaltose (Ferinject^®^ 50 mg/mL) or placebo	52.3% of patients in the ferric carboxymaltose group showed a 33% improvement in endurance time, compared to 18.2% in the placebo group (*p* = 0.009)
Grasmuk-Siegl E et al. [77]	RCT	4 weeks	Austria	11	63 ± 8	72%/28%	1000 mg of intravenous ferric carboxymaltose (Ferinject^®^)	The 6MWT distance increased by 34.7 ± 34.4 m (*p* = 0.011). VO_2_max increased by 1.87 ± 1.2 mL/kg/min (*p* = 0.006). The average SGRQ score decreased by 7.56 ± 6.12 units (*p* = 0.004)
Santer P et al. [78]	RCT	8 weeks	UK	48	69 ± 8	70%/30%	A single intravenous iron replacement (ferric carboxymaltose, FCM; 15 mg/kg body weight) or saline placebo	In the FCM group, 29.2% of participants showed a ≥40 m improvement in the 6MWD. mMRC: 33.3% vs. 66.7%, *p* = 0.02

Abbreviations: RCT: randomized controlled trial; MDA: malondialdehyde; GSH: glutathione; VO_2_max: maximal oxygen consumption; SGRQ: St. George’s respiratory questionnaire; 6MWT: 6 min walk test; FCM: ferric carboxymaltose; MMRC: modified medical research council.

**Table 3 nutrients-16-04118-t003:** Summary of RCTs investigating the effects of magnesium sulfate in COPD exacerbation.

Study	Design	Mean Follow-Up	Country	Sample Size	Average Age (Year)	Sex Male/Female	Intervention	Main Results
do Amaral AF et al. [79]	RCT	45 min	Brazil	22	64 ± 6	100% male	2 g of intravenous magnesium sulfate or placebo on two separate occasions	Functional vital capacity: −0.48 L (95% CI: −0.96, −0.01); Inspiratory capacity: 0.21 L (95% CI: 0.04, 0.37); Maximum inspiratory pressure: 10 cmH_2_O (95% CI: 1.6, 18.4); Maximum expiratory pressure: 10.7 cmH_2_O (95% CI: 0.20, 21.2)
Mukerji S et al. [80]	RCT	120 min	New Zealand	30	62 ± 4	50%/50%	Standard bronchodilator therapy and either placebo (saline) or 2 g of intravenous magnesium sulfate	At T120, the mean FEV_1_ change was 27.07% in the magnesium group, compared to 11.39% in the placebo group (95% CI: 3.7–27.7; *p* = 0.01)
Amaral AF et al. [81]	RCT	100 min	Iran	20	66.2 ± 8.3	70%/30%	2 g of magnesium sulfate or normal saline intravenously on two separate occasions	Magnesium infusion significantly decreased functional residual capacity (−0.41 L) and residual volume (−0.47 L), and increased maximum load (+8 W) and respiratory gas exchange ratio (+0.06) at peak exercise
Skorodin MS et al. [82]	RCT	45 min	USA	72	62.8 ± 9.0	97%/3%	Either 1.2 g of magnesium sulfate or a placebo was administered intravenously after nebulized albuterol	The magnesium sulfate group showed a significantly greater increase in peak expiratory flow (22.4% ± 28.5%) compared to the placebo group (6.1% ± 24.4%) (*p* = 0.01)
Vafadar Moradi E et al. [83]	RCT	90 min	Iran	77	-	-	The magnesium sulfate (MgSO_4_) group (MG) received 2.5 g of magnesium sulfate in 50 mL saline, while the placebo group received 5 mL of sterile water in 50 mL saline	The MG showed a significant increase in PEFR (15.67% ± 3.35) compared to the placebo group (5.03% ± 6.29). The MG also showed a significant improvement in DSS (−3.69 ± 1.07) compared to the placebo group (−2.05 ± 1.11)
Abreu González J et al. [84]	RCT	45 min	Spain	24	64 (57–78)	50%/50%	1.5 g of magnesium sulfate or placebo intravenously over 20 min on two separate days	The percentage increase in FEV_1_ was 17.11% (3.7%) after magnesium sulfate and 7.06% (1.8%) after placebo (*p* = 0.008)
Solooki M et al. [85]	RCT	3 day	Iran	30	68 ± 9	70%/30%	Group A received 2 g of magnesium sulfate in saline over 20 min for three days in addition to standard therapy, while Group B received standard medications	After 45 min, the FEV_1_ value was 27% ± 9 in Group A and 36% ± 20 in Group B (*p* = 0.122). On day three, the FEV_1_ value was 32% ± 17 in Group A and 41% ± 22 in Group B (*p* = 0.205)
Zanforlini BM et al. [86]	RCT	6 months	Italy	49	73.0 ± 8.9	76%/24%	300 mg of magnesium citrate daily	The intervention group had lower CRP levels than the placebo group (β = −3.2, *p* = 0.03)
Cömert Ş et al. [87]	RCT	120 min	Turkish	20	-	.	One group received 500 µg of ipratropium bromide (IB) and 151 mg of magnesium sulfate via inhalation, while the other group received IB with a placebo	At 10 min, the PEFR increase was significantly higher in the magnesium group (4.7 vs. −3.5, *p* = 0.005). At 30 min, the PEFR increase was also significantly higher in the magnesium group (8.2 vs. 1.3, *p* = 0.03)

Abbreviations: RCT: randomized controlled trial; FEV_1_: forced expiratory volume in 1 s; PEFR: peak expiratory flow rate; DSS: dyspnea severity scale; MgSO_4_: magnesium sulfate; CI: confidence interval; IB: ipratropium bromide.

**Table 4 nutrients-16-04118-t004:** The role of trace elements in the pathogenesis and management of COPD.

Trace Element	Biological Mechanisms	Pathogenesis in COPD	Therapeutic Relevance
Iron	Essential for hemoglobin and cytochrome function; supports oxygen transport and cellular respiration; key for antioxidant enzymes (e.g., catalase)	Deficiency: worsened hypoxia, increased oxidative stress. Excess: ROS generation via the Fenton reaction, promoting inflammation and oxidative damage	Altered serum iron levels during AECOPD are associated with dyspnea and reduced lung function
Magnesium	Crucial for ATP production, muscle relaxation, and cofactor for over 300 enzymes, including those in antioxidant defense	Deficiency: reduced bronchodilation, increased inflammation and oxidative stress, higher hospitalization rates	Magnesium supplementation improves bronchodilation and reduces exacerbation frequency
Zinc	Regulates immune response, antioxidant defense, and tissue repair; controls metalloproteinase activity	Deficiency: weakened immunity, increased susceptibility to infections, oxidative stress, impaired regeneration	Zinc supplementation reduces inflammation and oxidative damage
Calcium	Critical for muscle contraction, signal transduction, and release of inflammatory mediators	Deficiency: impaired diaphragm function, worsened bronchospasms	Calcium supplementation improves respiratory muscle performance
Selenium	Integral for antioxidant enzymes such as glutathione peroxidase	Deficiency: weakened antioxidant defense, oxidative damage, worse outcomes during AECOPD	Selenium supplementation enhances antioxidant capacity and reduces inflammation
Copper	Supports the function of Cu-Zn SOD antioxidant enzyme and acts as a cofactor for cytochrome-c oxidase	Deficiency: reduced antioxidant defense, increased susceptibility to infections	Maintaining adequate copper levels supports immune and antioxidant systems
Manganese	Essential for Mn-SOD antioxidant enzyme activity; protects mitochondria from oxidative stress	Deficiency: increased mitochondrial ROS production, lung tissue damage	Manganese supplementation improves mitochondrial function and reduces oxidative stress

Abbreviations: AECOPD: Acute exacerbations of chronic obstructive pulmonary disease; ROS: Reactive oxygen species; SOD: Superoxide dismutase.

**Table 5 nutrients-16-04118-t005:** Recommended daily intake of essential trace elements and their effects in COPD patients.

Trace Element	Recommended Daily Intake (NRV)	Effect in COPD Patients
Zinc	10–20 mg	Antioxidant and immune-supporting properties reduce inflammation and the frequency of exacerbations.
Iron	8–15 mg	Enhances antioxidant defense and respiratory support, improving quality of life.
Magnesium	300–500 mg	Supports respiratory muscle function; useful as an adjunct therapy to improve breathing.
Selenium	50–70 µg	Anti-inflammatory and tissue-protective antioxidant properties.
Manganese	1–2 mg	Reduces inflammation and protects tissues with antioxidant effects.
Copper	0.8–2.4 mg	Essential for antioxidant enzyme function and may serve as a biomarker for disease progression.
Calcium	1200–1500 mg	Vital for muscle strength and bone health, especially during long-term corticosteroid therapy.

Source: table compiled by the authors based on the findings presented in this review article. Further investigation is essential to establish an evidence-based protocol for mineral supplementation tailored to patients with COPD, optimizing clinical outcomes and addressing the disease’s complex nutritional needs. Note: Trace element supplementation requires careful monitoring to avoid toxicity and should be integrated into comprehensive COPD management strategies based on individual patient needs. NRV: Nutrient Reference Value.

## Data Availability

Data sharing is not applicable to this article as no new data were created or analyzed in this study.

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
