# Peer review of "The Role of Trace Elements in COPD: Pathogenetic Mechanisms and Therapeutic Potential of Zinc, Iron, Magnesium, Selenium, Manganese, Copper, and Calcium"

_nutrients, 2024, doi:10.3390/nu16234118_

Round 1
Reviewer 1 Report
Comments and Suggestions for Authors
Dear Dr Varga,
In my opinion, the review “The Role of Trace Elements in COPD Management: Focus on the Therapeutic Potential of Zinc, Iron, Magnesium, Selenium, Manganese, Copper, and Calcium” is an actual and opportune a contribution to the COPD research. In addition, the methodology of the database analysis is meticulous and appropriate, the style is clear and easily understandable. The only drawback I found, mentioned also in the “Limitations” chapter, is that the scientific quality of the study was not one of the inclusion or exclusion criteria.
Author Response
Dear Reviewer 1,
Thank you for your valuable comments, particularly your observations regarding the methodology and limitations. We fully agree that due to the nature of the narrative review, certain results or research areas may have been omitted from the analysis, and that the relationship between trace elements and COPD is complex and multifaceted.
In response to your comments regarding the limitations, we have expanded the manuscript with the following text in the Limitations section:
This review is not systematic; therefore, certain results or research areas may have been omitted from the analysis. The relationship between trace elements and COPD is complex and multifaceted, and not all studies meet the same quality standards, making it difficult to draw definitive conclusions. Additionally, methodological differences between studies, varying techniques used for measuring trace elements, and the different stages of COPD further increase the heterogeneity of the results.
A significant limitation is that the scientific quality of the studies included in the review was not an inclusion or exclusion criterion. While the narrative review encompasses a wide range of studies, including randomized controlled trials and observational studies, the variability in the methodological rigor of the included studies may influence the reliability and generalizability of the conclusions. Furthermore, some of the results are based on cross-sectional studies, which do not allow for the clear determination of causal relationships.
We hope that these modifications appropriately reflect the points raised in your critique and contribute to enhancing the scientific quality of the manuscript.
Best regards,
János T Varga, MD, PhD
Reviewer 2 Report
Comments and Suggestions for Authors
The review article looks too scattered and is not easy to read. The authors included too many trace elements within 1 article.
The background knowledge is definitely inadequate. Please elaborate the biological mechanisms in depth.
The authors discuss their effects in AECOPD as well as stable state which is confusing. The authors should reorganize the whole article, discuss the different roles in different settings.
Also, for Mg, it is a well recognized treatment for asthma AE. It should not be treated as the same as other trace elements. Please consider remove the part on Mg.
The author should consider adding diagrams to the review article.
The discussion session should focus on the findings they quoted. Pathogenesis of COPD and trace elements is NOT discussed in the article.
Comments on the Quality of English LanguageNA
Author Response
Dear Reviewer 2,
We sincerely thank you for your careful and insightful comments regarding our manuscript. We highly appreciate your feedback and are committed to improving the quality of the article based on your suggestions. Regarding your observation that the review appears fragmented and difficult to follow due to the inclusion of too many trace elements, we acknowledge this concern. However, the goal of this narrative, comprehensive review is to present as many relevant trace elements as possible within a single summary article. We have highlighted the inherent limitations of this approach, noting that narrative reviews encompass a wide range of studies, including randomized controlled trials and observational research. The variability in methodological rigor across these studies may influence the reliability and generalizability of the conclusions. Additionally, some findings are based on cross-sectional studies, which do not allow for definitive causal relationships to be established.
We are also grateful for your feedback regarding the lack of background information on biological mechanisms. In response, we have expanded our discussion to elaborate on how specific trace elements influence the pathophysiology of COPD, as detailed in both the introduction and discussion sections. Furthermore, we have revised the discussion section to provide a more in-depth exploration of the role of trace elements in COPD pathogenesis. We have also included an additional explanatory table in the discussion to enhance clarity. Regarding magnesium, we appreciate your comment on its established role in the treatment of acute asthma exacerbations. We revisited the decision to discuss magnesium alongside other trace elements in this review. After careful consideration, we have retained magnesium in this comprehensive review due to its unique role and importance, not only in asthma but also in COPD, where it serves as a valuable trace element. Due to the article’s length and space constraints, we opted not to add additional figures, diagrams, or tables, as we believe this would further increase the complexity of the manuscript. However, we have worked on improving readability and have also revised the abstract for better clarity and conciseness. We sincerely thank you for your valuable suggestions, and we have made the necessary revisions to the manuscript based on your recommendations.
Yours sincerely,
Prof. János T Varga
Reviewer 3 Report
Comments and Suggestions for Authors
I had to revise the article entitled ”The Role of Trace Elements in COPD Management: Focus on 2 the Therapeutic Potential of Zinc, Iron, Magnesium, Selenium, 3 Manganese, Copper, and Calcium” submitted for publishing in Nutrients journal.
The present review investigated the role of several micronutrients in improving clinical control or lung function in COPD. It is a narrative review that presents the results of the most important studies, including randomised controlled trials but also other types, lie cross-sectional studies. It is a comptrehensive narrative review but several improvements are needed in order to be accepted for publication
1. In the abstract there is a typing error
2. Method section – lines 92-94 should be excluded, the primary goal is already mentioned in the introduction section
3. I have the same advice for lines 100-103 in method section
4. In the inclusion criteria the authors included the parameter named exacerbation in lung function. My opinion is that should be separated by spirometry, it is a clinical one.,
5. Please revise all the tables. As main results you should briefly describe the main findings in each study not the entire description.
e.g Instead of The COPD patients received a daily supplement of 22 mg of zinc picolinate for eight weeks. You should mention 22 mg zinc picolinate, 8 weeks.
6. In the discussion section, lines 418-419, I suggest to rephrase …. “production or release of mediators”……, since it is known that release of mediators is a process controlled by intracellular calcium. Please provide a reference
7. Lines 469-498 – should be re-do since the same ideas are already presented before of are re-analyzed in the rest part of discussion
8. lines 516-517 – the authors should exclude them since nothing related to zinc transporters in COPD is mentioned previously in the results
9. Line 568 the authors mentioned that El-Attar et al reported that the infusion of selenium can reduce the duration of mechanical ventilation, but in that study several trace elements were given. Please revise the phrase.
10. table 4 includes some recommendation. These are recommendation of the authors of the article based on the findings from different studies or there are included in guidelines? They should explain and include references.
Author Response
Dear Reviewer 3,
Thank you very much for your valuable feedback. We have revised the abstract and methodology sections, removing lines 92–94 and 100–103 to avoid repetition. We have also separated spirometry measurements and the number of exacerbations under the inclusion criteria. The tables have been improved and shortened by simplifying the descriptions of interventions and outcomes wherever possible. Additionally, we have rephrased lines 418 and 419, including appropriate references, and revised lines 469–498, supplementing the discussion with a new table. The zinc transporters have been removed from the manuscript, and the study by El-Attar has been corrected in the discussion section.
There are currently no specific recommendations for trace element supplementation tailored to COPD patients for each individual element. The recommendations presented in our manuscript are based on the results of various studies, compiled into a custom-designed table, which is clearly indicated beneath the table. Regarding precise supplementation, it must always be tailored to the needs of individual patients, and this note has also been added under the table. Thank you once again for your helpful comments.
Yours sincerely,
Prof. János T Varga
Reviewer 4 Report
Comments and Suggestions for Authors
Line 60-61
Please indicate whether the development of COPD is associated with an increase or decrease in the levels of these elements.
Line 106-107
What consideration is given to this age requirement?
Line 224-225
How do copper, selenium, and zinc levels relate to the development of COPD? Is there a relationship between the three elements? This study did not include supplementation of these elements?
Line 262-265
Is this therapeutic impact limited to a certain demographic of COPD patients? Are there any limitations?
Line 272
What is the specific role of magnesium sulfate? Anti-inflammatory or antioxidant effects, etc.?
Line 339-342
What is the general relationship between antioxidants and lung function, and how does this relationship change depending on the patient's health state or illness severity?
Line 358-361
Do these elements have opposite effects on COPD patients? Are there any interactions or synergistic effects while utilizing multiple elements simultaneously?
Line 400-403
What are the probable causes of elevated levels of copper and zinc levels in COPD patients?
Line 458-460
There is no strong evidence that vitamin D supplementation slows the decline in lung function.
Line 515-517
Are there good biomarkers for zinc status in the body?
Line 572-574
Mechanisms underlying selenium’s role in COPD are wanted to elucidate this association further.
Line 606-608
Is it clear that their clinical efficacy is related to the pharmacological modulation of the oxidant/antioxidant balance?
Line 641-643
Highlight molecular targets that are worth exploring.
Author Response
Dear Reviewer 4,
Thank you for your valuable feedback. We have supplemented the introduction to clarify the role of trace elements in the pathogenesis of COPD, which we have highlighted in red. COPD typically occurs in individuals over 40 years of age, as the disease develops due to long-term exposure to risk factors such as smoking and air pollution. The age requirement ensures that the studied population is genuinely affected by COPD. Furthermore, comorbidities and physiological changes that influence symptoms or treatment responses are more common in older populations. Defining an age threshold enhances the interpretability of results, as younger patients with COPD are rare and often have underlying conditions (e.g., genetic disorders such as alpha-1 antitrypsin deficiency) that may not align with the study's objectives. This criterion ensures that findings are applicable to the majority of COPD patients, who are typically over 40, thereby supporting relevant and reliable outcomes.
As requested, we expanded on the study by Koç U. et al., addressing the relationship between copper, selenium, and zinc in COPD. These trace elements are critical in antioxidant defense, a key factor in slowing COPD progression. We also added limitations and clarified findings in lines 224–225.
Magnesium plays an essential role with its anti-inflammatory properties and is vital for muscle function, including respiratory muscles, as discussed in detail in the manuscript. Among COPD patients with magnesium deficiency, higher levels of systemic inflammation and reduced muscle strength are observed, worsening disease outcomes. Magnesium supplementation improves inflammation and muscle function, thereby enhancing patients' quality of life.
We revised line 272 to include:
Magnesium sulfate acts through multiple mechanisms in the management of COPD exacerbations. Its calcium channel-blocking properties reduce smooth muscle contraction, while inhibiting acetylcholine release relaxes bronchial smooth muscles. Additionally, it has anti-inflammatory effects by decreasing airway neutrophil activity and histamine release from mast cells. Intravenous magnesium sulfate improves lung function and respiratory rate, especially when combined with other bronchodilator therapies.
We supplemented line 339 with:
Antioxidants are crucial in COPD as they neutralize free radicals that cause oxidative stress in lung tissues. Selenium, for instance, protects against inflammation and tissue damage. This relationship is more pronounced in severe disease cases, where chronic lung conditions like COPD exacerbate oxidative stress, making antioxidant interventions vital for treatment and lung function improvement.
Line 358 now includes: When multiple trace elements such as selenium, manganese, zinc, and copper are co-administered, synergistic effects may occur. For instance, manganese and selenium jointly activate antioxidant enzymes such as glutathione peroxidase and superoxide dismutase. This enhances antioxidant protection and reduces oxidative stress, a critical factor in treating inflammatory conditions like COPD.
Line 400 was revised to state:
Elevated levels of copper and zinc in COPD may reflect oxidative stress and inflammatory responses associated with the disease, which intensify during progression and exacerbations.
Line 458 was reformulated and marked in red:
Although biomarkers are available for assessing zinc levels, there is no single ideal biomarker that reliably reflects zinc status under all circumstances. Plasma or serum zinc concentration is commonly used but can be influenced by numerous factors, making it less reflective of long-term zinc status. Measuring urinary zinc levels and zinc transporter proteins such as ZnT-2 can also be useful, but their interpretation requires context as they are affected by other variables.
We elaborated on the therapeutic mechanisms of selenium, explaining its role in selenoproteins with antioxidant and enzymatic functions. Selenium integrates into various proteins to form selenoproteins, which play key roles in reducing oxidative stress and inflammation—common pathophysiological factors in COPD. Specifically, selenoproteins such as glutathione peroxidases and thioredoxin reductases help mitigate oxidative damage, protecting lung tissues and reducing systemic inflammation. Selenium's antioxidant and anti-inflammatory effects improve patient outcomes, particularly in preventing exacerbations and supporting rehabilitation. However, further clinical studies are needed to determine the optimal dosage and treatment regimens for patients.
Finally, we emphasized molecular targets worthy of further investigation at the end of the article.
Thank you again for your valuable review.
Sincerely,
János T Varga, MD, PhD